# Iron Overload-Related Oxidative Stress Leads to Hyperphosphorylation and Altered Anion Exchanger 1 (Band 3) Function in Erythrocytes from Subjects with β-Thalassemia Minor

**DOI:** 10.3390/ijms26041593

**Published:** 2025-02-13

**Authors:** Sara Spinelli, Elisabetta Straface, Lucrezia Gambardella, Daniele Caruso, Silvia Dossena, Angela Marino, Rossana Morabito, Alessia Remigante

**Affiliations:** 1Department of Chemical, Biological, Pharmaceutical and Environmental Sciences, University of Messina, 98166 Messina, Italy; saspinelli@unime.it (S.S.); marinoa@unime.it (A.M.); rmorabito@unime.it (R.M.); 2Biomarkers Unit, Center for Gender-Specific Medicine, Istituto Superiore di Sanità, 00161 Rome, Italy; elisabetta.straface@iss.it (E.S.);; 3Complex Operational Unit of Clinical Pathology of Papardo Hospital, 98166 Messina, Italy; caruso.daniele1985@libero.it; 4Institute of Pharmacology and Toxicology, Research and Innovation Center Regenerative Medicine & Novel Therapies, Paracelsus Medical University, 5020 Salzburg, Austria; silvia.dossena@pmu.ac.at; 5Department of Biomedical, Dental and Morphological and Functional Imaging, University of Messina, 98166 Messina, Italy

**Keywords:** β-thalassemia minor (β-Thal^+^), oxidative stress, anion exchanger 1 (AE1), Na^+^-K^+^/ATPase, red blood cells

## Abstract

β-thalassemia, a hereditary hemoglobinopathy, is caused by reduced or absent synthesis of the β-globin chains of hemoglobin. Three clinical conditions are recognized: β-thalassemia major, β-thalassemia intermedia, and β-thalassemia minor (β-Thal^+^). This latter condition occurs when an individual inherits a mutated β-globin gene from one parent. In erythrocytes from β-Thal^+^ subjects, the excess α-globin chains produce unstable α-tetramers, which can induce substantial oxidative stress leading to plasma membrane and cytoskeleton damage, as well as deranged cellular function. In the present study, we hypothesized that increased oxidative stress might lead to structural rearrangements in erythrocytes from β-Thal^+^ volunteers and functional alterations of ion transport proteins, including band 3 protein. The data obtained showed significant modifications of the cellular shape in erythrocytes from β-Thal^+^ subjects. In particular, a significantly increased number of elliptocytes was observed. Interestingly, iron overload, detected in erythrocytes from β-Thal^+^ subjects, provoked a significant production of reactive oxygen species (ROS), overactivation of the endogenous antioxidant enzymes catalase and superoxide dismutase, and glutathione depletion, resulting in (a) increased lipid peroxidation, (b) protein sulfhydryl group (-SH) oxidation. Iron overload-related oxidative stress affected Na^+^/K^+^-ATPase activity, which in turn may have contributed to impaired β-Thal^+^ erythrocyte deformability. As a result, alterations in the distribution of cytoskeletal proteins, including α/β-spectrin, protein 4.1, and α-actin, in erythrocytes from β-Thal^+^ subjects have been detected. Significantly, oxidative stress was also associated with increased phosphorylation and altered band 3 ion transport activity, as well as increased oxidized hemoglobin, which led to abnormal clustering and redistribution of band 3 on the plasma membrane. Taken together, these findings contribute to elucidating potential oxidative stress-related perturbations of ion transporters and associated cytoskeletal proteins, which may affect erythrocyte and systemic homeostasis in β-Thal^+^ subjects.

## 1. Introduction

Thalassemias are a heterogeneous group of hereditary hemoglobinopathies characterized by the autosomal recessive inheritance of defects in the production of the globin chains of hemoglobin [1]. Hemoglobin protein is a hetero-tetramer composed of two α-chain subunits and two β-chain subunits [2]. The clinical manifestations of thalassemia syndrome range from an asymptomatic state without complications to severe transfusion-dependent anemia with prenatal onset [3]. Historically, the prevalence of thalassemia has been highest in the Mediterranean area [4]. However, as a result of mass migrations of populations from high-prevalence areas, thalassemias are now encountered in most countries, including North Europe, the United States, Canada, South America, and Australia [5]. Thalassemias can be distinguished according to the defective globin chain, and the clinically important forms are α-thalassemia and β-thalassemia [6]. A-thalassemia is characterized by a reduced synthesis of α-globin chains, while β-thalassemia results from mutations in one or both of the β-globin genes on the short arm of chromosome 11, which lead to inadequate production or lack of production of the β-globin chains. Β-thalassemia includes three main forms: thalassemia major, also known as Mediterranean anemia, thalassemia intermedia, and thalassemia minor [7].

Beta-thalassemia minor (β-Thal^+^) occurs when an individual inherits a mutated β-globin gene from one parent, leading to mild anemia [6]. In erythrocytes from β-Thal^+^ subjects, the excess α-globin chains produce unstable α-tetramers, which can induce substantial oxidative stress leading to plasma membrane and cytoskeleton damage, as well as deranged cellular function [8]. In addition, α-globin precipitates can also cause the production of abnormally shaped erythrocytes in the bloodstream. In fact, examination of a peripheral blood smear obtained from a β-Thal^+^ subject may show microcytosis and, more frequently, anisocytosis and poikilocytosis [9]. The main causes of oxidative stress in β-Thal^+^ erythrocytes are the degradation of the unstable hemoglobin and iron overload, both stimulating the production of excess reactive species (ROS) [10,11]. ROS could directly target membrane lipids containing carbon–carbon double bond(s), especially polyunsaturated fatty acids (PUFAs), and -SH groups of proteins [12,13,14,15]. Such ROS elevation, when potentially combined with decreased endogenous antioxidant capacity, may further lead to functional defects in both membrane lipids and cytoskeletal proteins [16,17].

The membrane skeleton structure of erythrocytes is a network structure composed of proteins, especially spectrin, that maintains the geometry of the cells by folding and unfolding under the influence of different mechanical stress [18]. This spectrin-based network connects with the membrane lipid matrix via integral proteins, including band 3 protein, actin, and through junctional protein band 4.1 [19]. The alterations in the interactions of the cytoskeleton with the plasma membrane proteins could reduce erythrocyte deformability, which in turn could affect the effective delivery of oxygen to target tissues [19,20]. In this regard, anion exchanger 1 (AE1/*SLC4A1*) or band 3 protein, the predominant glycoprotein of the erythrocyte membrane, promotes the chloride/bicarbonate (Cl^−^/HCO_3_^2−^) anion exchange across the plasma membrane, a process necessary for efficient respiration [21]. The mechanisms underlying deranged tissue oxygenation might include an altered anion exchange capability due to increased oxidative stress. In addition, it has been widely reported that oxidized band 3 is phosphorylated by Syk kinase at tyrosine 8 and 21 [22]. In this context, band 3 protein phosphorylation can lead to conformational modifications of protein structure and modulate protein–protein interactions.

The typical biconcave shape can provide erythrocytes with a greater area available for gas exchange and, in parallel, ensures adaption to the narrow capillaries and efficient tissue oxygenation. However, the production of ROS may seriously impair erythrocyte deformability by auto-oxidation, causing membrane phospholipid peroxidation, protein degradation, methemoglobin formation, and subsequent hemolysis [19]. The impaired deformability of circulating erythrocytes affects systemic microcirculation and may cause profound tissue hypoxia [19,23]. Another factor regulating erythrocyte deformability is Na^+^/K^+^-ATPase activity. Localized in the erythrocyte membrane, the Na^+^/K^+^-ATPase pump maintains the optimal intracellular cationic concentrations, thus regulating both cellular volume and water homeostasis and, consequently, the surface area-to-volume ratio [24].

During the clinical course of β-Thal^+^, changes in blood rheological parameters, such as a decrease in erythrocyte membrane elasticity (reduced deformability) and surface-to-volume ratio or a rise in viscosity, which can reflect in deranged tissue oxygenation, are commonly encountered [25]. In β-Thal^+^, the rheological properties of erythrocytes can be impaired as a result of defects in cellular membrane skeletal architecture.

The relationship between the increased oxidative stress in β-Thal^+^ erythrocytes and their abnormal rheological properties is still poorly documented. In particular, both band 3 protein phosphorylation and rearrangements in cytoskeleton proteins might contribute to reduced cellular deformability. Thus, the aim of the present work was to investigate structural rearrangements, including deformability, associated with an increased oxidative stress potentially provoked by an iron overload in erythrocytes from β-Thal^+^ volunteers. Specifically, we hypothesized that increased oxidative stress may lead to band 3 protein modifications and the alteration of ion transport activity. To verify this hypothesis, the anion exchange capability, cellular distribution, and phosphorylation of band 3 protein, as well as the activity of the endogenous antioxidant system, were ascertained.

## 2. Results

### 2.1.β-. Thal^+^ Erythrocyte Shape and Plasticity

As reported in Figure 1A, morphological changes were observed in human erythrocytes from β-Thal^+^ volunteers compared to cells from healthy volunteers. In particular, elliptocyte cells (red arrows), namely cells with an oval or elongated shape, are recognizable. The percentage of morphologically altered cells in blood samples obtained from β-Thal^+^ volunteers was less than 1%.

For each shear stress value, the deformability (Eis) of cells obtained from β-Thal^+^ volunteers was significantly lower than that of erythrocytes obtained from healthy volunteers (Figure 1B). The EI max for β-Thal^+^ erythrocyte samples was 0.54 compared to 0.64 for control cells (erythrocytes from healthy volunteers).

### 2.2. Evaluation of Ion Transport in β-Thal+ Erythrocytes

#### 2.2.1. Na^+^/K^+^ ATPase Activity

In erythrocytes from β-Thal^+^ volunteers, the activity of the Na^+^/K^+^ ATPase pump was significantly higher than that detected in cells from healthy volunteers (Figure 2). In parallel, after the treatment of cells from healthy volunteers with a pro-oxidant compound (50 mM of AAPH for 1 h, at 37 °C), the Na^+^/K^+^ ATPase activity significantly increased compared with untreated cells.

#### 2.2.2. SO_4_^2−^ Uptake via Band 3 Protein

To evaluate the anion exchange capability of band 3 protein, the SO_4_^2−^ uptake was determined in erythrocytes obtained from healthy and β-Thal^+^ volunteers (Figure 3). In control cells, SO_4_^2−^ uptake progressively increased, reaching equilibrium in 17.79 min (rate constant of SO_4_^2−^ uptake is 0.056 ± 0.010 min^−1^; Table 1). In cells from β-Thal^+^ volunteers, the transport rate constant (0.115 ± 0.021 min^−1^; Table 1) was higher than that detected in cells from healthy volunteers, thus denoting accelerated transport kinetics. In both experimental conditions, no significant difference in SO_4_^2−^ amount internalized by erythrocytes after 45 min of incubation in SO_4_^2−^ medium was reported (Table 1). In DIDS-treated cells, the rate constant of SO_4_^2−^ uptake and the SO_4_^2−^ amount internalized were substantially reduced compared to control cells.

### 2.3. Oxidative Stress in β-Thal^+^ Erythrocytes

#### 2.3.1. Detection of ROS Production

In Figure 4A, intracellular ROS levels were measured in healthy and β-Thal^+^ volunteers or, alternatively, in cells from healthy volunteers exposed to 20 mM H_2_O_2_ (for 30 min, at 25 °C). As expected, the treatment of cells with 20 mM H_2_O_2_ significantly increased ROS levels. Importantly, in erythrocytes obtained from β-Thal^+^ volunteers, ROS levels were found to be significantly elevated compared to cells obtained from healthy volunteers.

#### 2.3.2. Detection of TBARS Levels

Figure 4B reports the TBARS levels in cells obtained from healthy and β-Thal^+^ volunteers. As expected, after the treatment of cells from healthy volunteers with the pro-oxidant AAPH (50 mM for 1 h, at 37 °C), the TBARS levels were significantly higher than those detected in untreated cells. Similarly, in the samples obtained from β-Thal^+^ volunteers, the TBARS levels were significantly higher than those measured in erythrocyte samples from healthy volunteers.

#### 2.3.3. Detection of Total -SH Groups

Figure 4C shows the total content of sulfhydryl groups in human erythrocytes obtained from healthy and β-Thal^+^ volunteers. As expected, incubation with NEM, used as a positive control (2 mM for 1 h, at 25 °C), led to a significant reduction in the sulfhydryl groups content compared to erythrocytes from healthy and β-Thal^+^ volunteers. The sulfhydryl group content detected in samples obtained from β-Thal^+^ volunteers was significantly lower than that measured in cells obtained from healthy volunteers.

#### 2.3.4. Evaluation of the Endogenous Antioxidant Capacity

SOD and CAT activity, as well as the GSH/GSSG ratio, were measured as an estimate of the endogenous antioxidant capacity of the human erythrocytes obtained from both healthy and β-Thal^+^ volunteers (Figure 5). As expected, the treatment of cells with 20 mM H_2_O_2_ for 30 min at 25 °C significantly increased both SOD and CAT activity. Parallelly, exposure to H_2_O_2_ also led to a decreased GSH/GSSG ratio, reflecting increased GSSG levels and/or decreased GSH levels. In erythrocyte samples obtained from β-Thal^+^ volunteers, the activities of both SOD and CAT were significantly higher than those of samples obtained from healthy volunteers, and the GSH/GSSG ratio was decreased (Figure 5).

### 2.4. Detection of MetHb Levels and Intracellular Iron Release in β-Thal^+^ Erythrocytes

Figure 6A reports MetHb levels detected in human erythrocytes obtained from healthy and β-Thal^+^ volunteers. The methemoglobin levels measured after exposure of cells from healthy volunteers to a well-known MetHb-forming compound (NaNO_2_, 4 mM for 1 h, at 25 °C) were significantly higher than those detected in untreated cells. MetHb levels measured in samples from β-Thal^+^ volunteers were also significantly higher than those measured in samples obtained from healthy volunteers. In parallel, the amount of intracellular free iron release is reported in Figure 6B. In cells obtained from β-Thal^+^ volunteers, intracellular free iron levels were higher than those detected in control erythrocytes.

### 2.5. Assessment of Membrane and Cytoskeletal Protein Levels and Distribution in β-Thal^+^ Erythrocytes

#### 2.5.1. Band 3 Expression Levels and Phosphorylation

Figure 7A shows the band 3 protein expression levels detected in erythrocyte membranes obtained from healthy and β-Thal^+^ volunteers. In both conditions, no significant difference in band 3 protein expression levels has been reported. On the contrary, an increase in band 3 protein phosphorylation levels in cells from β-Thal^+^ volunteers has been detected (Figure 7B).

#### 2.5.2. Distribution of Band 3 Protein

The distribution of band 3 protein was evaluated by flow cytometry and immunofluorescence analysis in cells obtained from healthy and β-Thal^+^ volunteers. In samples obtained from β-Thal^+^ volunteers, the band 3 fluorescence signal was significantly higher than that of samples obtained from healthy volunteers (Figure 8A). Figure 8B shows an intense rearrangement in large clusters and redistribution towards the cell membrane (red arrows) of band 3 in cells obtained from β-Thal^+^ volunteers. An abnormally shaped cell, referred to as an elliptocyte, is also shown.

#### 2.5.3. Distribution of α/β-Spectrin

The distribution of α/β-spectrin was evaluated by flow cytometry and immunofluorescence analysis in cells obtained from healthy and β-Thal^+^ volunteers. In erythrocyte samples obtained from β-Thal^+^ volunteers, the fluorescence signal of α/β-spectrin was significantly higher than that of samples obtained from healthy volunteers (Figure 9A). Figure 9B shows an intense rearrangement and redistribution with the formation of peripheral clusters (red arrows) of α/β-spectrin detected in cells obtained from β-Thal^+^ volunteers.

#### 2.5.4. Distribution of Band 4.1 Protein

The distribution of band 4.1 protein was evaluated by flow cytometry and immunofluorescence analysis in cells obtained from healthy and β-Thal^+^ volunteers. In erythrocyte samples obtained from β-Thal^+^ volunteers, the fluorescent signal of band 4.1 protein was significantly higher than that of samples obtained from healthy volunteers (Figure 10A). Figure 10B shows an intense rearrangement from a punctate pattern to large clusters and redistribution towards the cell periphery (red arrows) of band 4.1 protein in cells obtained from β-Thal^+^ volunteers.

#### 2.5.5. Distribution of α-Actin

The distribution of α-actin was evaluated by flow cytometry and immunofluorescence analysis in cells obtained from healthy and β-Thal^+^ volunteers. In erythrocyte samples obtained from β-Thal^+^ volunteers, the fluorescence signal of α-actin was significantly higher than that of samples obtained from healthy volunteers (Figure 11A). Figure 11B shows an intense rearrangement and redistribution in large clusters (red arrows) of α-actin in cells obtained from β-Thal^+^ volunteers.

## 3. Discussion

Degradation of the unstable hemoglobin and iron overload represent the leading causes of oxidative damage in erythrocyte cells from β-Thal^+^ subjects [26]. The subsequent alterations at the level of biochemical components of the erythrocyte plasma membrane can then compromise their morphological integrity and potentially increase their susceptibility to oxidative damage, resulting in a decrease in their survival in the bloodstream [27]. As a result, oxidative changes may reduce the ability of erythrocytes to deform in the blood circulation, impeding them from squeezing through narrow capillaries. Typically, the erythrocyte diameter is larger than the diameter of the capillaries, and a reduction in deformability impairs the ability of erythrocytes to deliver oxygen to the tissues [28]. Based on these considerations, much attention has been paid to structural modifications associated with cell shape in this work. In particular, the scanning electron microscopy (SEM) technique revealed a dual cell population in the β-Thal^+^ samples, composed of cells with a canonical biconcave shape and oval-shaped cells (elliptocytes) (Figure 1A). Any changes in erythrocyte shape could lead to a decreased plasma membrane asymmetry and, consequently, altered cellular deformability [27,29]. As is known, plasma membrane structural integrity depends on an asymmetrical arrangement of phospholipids between the inner and outlet leaflets. In case of disruption of this lipidic asymmetry, the consequent phosphatidylserine exposure on the outer leaflet leads to the rapid and controlled removal of the affected cells from systemic circulation before the cell damage may cause uncontrolled hemolysis [30]. Although a substantial number of cells showed an abnormal cellular shape (Figure 1A), probably associated with plasma membrane rearrangement, the percentage of eryptotic cells remained low. The increased fraction of ROS detected in the cells obtained from β-Thal^+^ volunteers (Figure 4A) may have reacted with plasma membrane lipids, increasing their peroxidation (Figure 4B), which in turn may have affected the plasma membrane structure. Accordingly, reduced erythrocyte deformability, represented by a reduced elongation index, has been found in erythrocytes obtained from β-Thal^+^ subjects (Figure 1B). These findings suggest that damage to membrane lipids is responsible for the impaired cellular deformability associated with increased oxidative stress.

A further feature related to erythrocyte shape deals with Na^+^/K^+^-ATPase. This ion pump maintains the optimal intracellular concentrations of cations to regulate erythrocyte volume, water homeostasis, and the surface area-to-volume ratio, with consequences for blood rheology [31]. Thus, any alterations in Na^+^/K^+^-ATPase activity could lead to a decrease in RBC deformability in the blood flow [32]. The structure of Na^+^/K^+^-ATPase is organized in two different subunits: (1) the α (isoforms 1, 2, and 3) catalytic subunit, which is responsible for the cleavage of ATP and features the binding sites for Na^+^ and K^+^; (2) the β-subunit, required for the insertion of the Na^+^/K^+^-ATPase into the plasma membrane [33]. To better investigate the role of Na^+^/K^+^-ATPase in maintaining erythrocyte deformability in β-Thal^+^ conditions, its activity has been explored. Data obtained in this regard revealed a higher Na^+^/K^+^-ATPase activity and, in parallel, an iron overload in the cells of β-Thal^+^ subjects (Figure 2 and Figure 6B). Although most of the data from the literature describe an inhibition of the Na^+^/K^+^-ATPase pump associated with increased iron content in nucleated cells, the stimulation of Na^+^/K^+^-ATPase by intracellular free iron seems specific for erythrocytes. Since human erythrocytes do not express the α2 isoform of the enzyme, it can be suggested that intracellular free iron may modify the α3 subunit via oxidation of -SH groups (Figure 4C), which can have contributed to the activation of the ion pump [33,34].

It is well known that iron is essential for metabolic processes, but its overload can lead to the production of excess reactive species, including superoxide anion (O_2_∙), hydrogen peroxide (H_2_O_2_), and hydroxyl radical (OH∙), through the Fenton and Haber–Weiss reactions [35,36]. The ROS production by intracellular free iron can directly oxidize the plasma membrane lipids and attack the –SH groups of the Na^+^/K^+^-ATPase enzyme. This can modulate its activity and lead to the observed increase in the release of inorganic phosphate (Pi) (Figure 2).

Hemoglobin can be one of the dominant factors leading to oxidative stress within the erythrocytes of β-Thal^+^ subjects [36]. Physiologically, human erythrocytes contain ∼3% of methemoglobin, as NADH-dependent cytochrome reductase effectively converts methemoglobin (Fe^3+^) to Fe^2+^-hemoglobin [37]. However, when oxidative stress levels become excessively high, ROS production exceeds the capacity of the intracellular antioxidant systems [38]. Therefore, it is likely that in cells from β-Thal^+^ subjects, the increase in ROS levels caused the oxidation of hemoglobin to methemoglobin (Figure 6A), thus increasing both erythrocyte susceptibility to premature oxidative damage and inhibiting physiological functions. In this context, our results also showed that the oxidation of hemoglobin was associated with a rearrangement of the distribution of band 3 in multiple clusters, possibly following the formation of dimers/oligomers (Figure 8B), with no reduction in band 3 protein expression levels (Figure 7A).

Band 3 protein is an important erythrocyte membrane-spanning protein involved in protein–protein interactions [21,39]. In response to oxidative stress, an abnormal number of proteins can be phosphorylated, including band 3 protein [40,41]. In this regard, it has already been demonstrated that band 3 phosphorylation affects spectrin/band 3 binding and provokes severe changes in the deformability and shape of human erythrocytes [42]. Here, we show that the increase in oxidative stress levels in cells from β-Thal^+^ volunteers induced intense phosphorylation of band 3 (Figure 7). The band 3 cytoplasmatic domain oxidation can activate the Syk docking to band 3 and inhibit tyrosine phosphatases [43,44]. Then, band 3 hyper-phosphorylation can result in relevant alterations in erythrocyte features, such as (1) the disruption of the band 3 ankyrin-bound form that connects spectrin to the plasma membrane (structural modification) and/or (2) anion transport alterations via band 3 (functional modification); both are determinants of efficient tissue oxygenation. In fact, band 3 is known to be involved in the chloride/bicarbonate (Cl^−^ or HCO_3_^−^) exchange across the plasma membrane [21,45] and its function can be evaluated by measuring the rate constant for SO_4_^2−^ uptake [46,47,48,49,50,51,52]. With this methodological approach, the anion exchange is slower and more easily measurable than the physiological uptake of Cl^−^ or HCO_3_^−^ [53,54,55]. As the mechanisms of deranged tissue oxygenation might be caused by an altered anion exchange capability mediated by band 3 protein in β-Thal^+^ subjects, SO_4_^2−^ uptake through the protein was measured. The present data reported that, in β-Thal^+^ erythrocytes, the rate constant for SO_4_^2−^ uptake was accelerated compared to control cells (Figure 3, Table 1). This finding was consistent with previous observations from our research group, pointing out that an increase in post-translation modifications such as oxidation or phosphorylation at the level of band 3 protein can alter its function of anion exchange in human erythrocytes [46,50,56].

Any damage to membrane and/or cytoskeletal proteins induced by oxidative stress can contribute to the impairment of erythrocyte deformability [57]. The cytoskeleton structure of the human erythrocyte is a complex structure organized by several proteins. α- and β-spectrin contribute to the maintenance of erythrocyte physiological structure, possibly affected by mechanical stress in the bloodstream [58,59,60,61]. The network of α- and β-spectrin binds to the lipid membrane matrix through integral proteins, including band 3 protein [62]. In this context, both the redistribution and clustering of α- and β-spectrin on the plasma membrane of erythrocytes from β-Thal^+^ subjects have been demonstrated (Figure 9A,B). These results are consistent with the presence of erythrocytes with an atypical shape (Figure 1B). In addition, the cytoskeleton structure is anchored to the erythrocyte plasma membrane at two specific points to give more stability and modulate cellular deformability under shear stress. The first point is the network assembly of α- and β-spectrin, protein 4.1, and α-actin (junctional complex); the second point includes the band 3/ankyrin complex [63]. Regarding protein 4.1 and α-actin, a significant increase in clustered proteins and their abnormal redistribution on the plasma membrane of β-Thal^+^ cells compared to control cells was found (Figure 10 and Figure 11). In summary, these findings confirm the interdependent relationship between cytoskeleton and erythrocyte deformability. This flexible and elastic network is able to stabilize the membrane bilayer without compromising its deformability, thus enabling the erythrocytes to withstand shear stress during their passage via the vascular system to guarantee their main function, that is, oxygen transfer to cells and body tissues [64].

Human erythrocytes are also involved in redox regulation. The antioxidant system of these cells is particularly effective and is based on both enzymatic and non-enzymatic mechanisms. Moreover, erythrocytes are able to transport high concentrations of antioxidant molecules, including glutathione (GSH) [65]. In this context, we investigated the efficiency of the principal endogenous antioxidant enzymes, namely CAT and SOD, in cells from healthy and β-Thal^+^ volunteers (Figure 5A,B). In particular, both SOD and CAT activity, measured in β-Thal^+^ cells, were higher than in control cells, which could reflect the activation of the endogenous antioxidant defense system to reduce the increase in ROS levels (Figure 4A). However, the increased SOD and CAT activity failed to compensate for the increase in ROS levels, as reflected by the increased generation of methemoglobin (Figure 6A). Then, CAT and SOD activity upregulation with concomitant elevation of ROS and methemoglobin content might reflect exhaustion of the intracellular antioxidant machinery. In addition to enzymatic antioxidants, human erythrocytes also possess non-enzymatic antioxidants, such as reduced GSH. GSH is a tripeptide with highly reactive sulfhydryl groups that may act non-enzymatically as a free radical acceptor to counteract oxidative damage [66,67]. The formation of methemoglobin measured in β-Thal^+^ cells (Figure 6A) could also derive from the depletion of GSH, which makes cells more prone to oxidative injury. Accordingly, the GSH/GSSG ratio was reduced in erythrocytes from β-Thal^+^ volunteers (Figure 5C). NADPH generated during the pentose phosphate pathway (PPP) is needed to preserve both reduced GSH and CAT activity [68]. Thus, defects in the PPP could render human erythrocytes susceptible to oxidative injury [69]. In this regard, the abnormal distribution of band 3 protein detected in β-Thal^+^ erythrocytes (Figure 8B) could also have been caused by a failure of PPP activation, with concurrent defects in glutathione recycling (Figure 5C) and evidence of increasing oxidative stress (Figure 4A).

## 4. Materials and Methods

### 4.1. Solutions and Chemicals for Erythrocyte Sample Processing

All chemicals were purchased from Sigma (Milan, Italy). Regarding stock solutions, 4,4′-diisothiocyanatostilbene-2,2′-disulfonate (DIDS, 10 mM) was dissolved in dimethyl sulfoxide (DMSO); N-ethylmaleimide (NEM, 310 mM) was dissolved in ethanol. The H_2_O_2_ experimental solution was obtained by diluting a 30% *v*/*v* stock solution in distilled water, whereas 2,2′-Azobis (2-methylpropionamidine) dihydrochloride (AAPH, 0.5 M) was dissolved in PBS. Neither ethanol nor DMSO ever exceeded 0.001% *v*/*v* in the experimental solutions and were previously tested on human erythrocytes to exclude possible hemolytic damage.

### 4.2. Preparation of Human Erythrocyte Samples

In this study, 44 volunteers (21 males and 23 females) with β-thalassemic minor (β-Thal^+^) and 53 healthy volunteers (28 males and 25 females) were enrolled. Venous blood from *n* = 97 volunteer blood donors was collected into ethylenediaminetetraacetic acid (EDTA) tubes. The two donor groups, namely healthy and β-thalassemic minor (β-Thal^+^) volunteers, exhibited typical hematological differences between them but minimal baseline variation in sex, age, and donation frequency (Table 2). β-Thal^+^ was confirmed by hemoglobin (Hb) electrophoresis (Hb A_2_ = 3.5–5%) and molecular identification of mutations. The study was approved by the Institutional Review Board (or Ethics Committee) of the University of Messina (prot. 52-22). Each investigation was carried out upon donor consent in accordance with the principles of the Declaration of Helsinki. Human erythrocytes were washed in isotonic solution (NaCl 150 mM, 4-(2-hydroxyethyl)-1-piperazineethanesulfonic acid (HEPES) 5 mM, glucose 5 mM, pH 7.4, osmolarity 300 mOsm/kgH_2_O) and centrifuged (Neya 16R, 1200× *g*, 5 min) to discard plasma and buffy coat. Finally, human erythrocytes were suspended in isotonic solution according to the experimental tests described below.

### 4.3. Analysis of Cell Shape by Scanning Electron Microscope (SEM)

Analysis of cell shape was performed in human erythrocyte samples from healthy and β-Thal^+^ volunteers. Cell samples were collected, plated on poly-l-lysine-coated slides, and fixed with 2.5% glutaraldehyde in 0.1 M cacodylate buffer (pH 7.4) at room temperature for 20 min. Then, samples were post-fixed with 1% OsO_4_ in 0.1 M sodium cacodylate buffer and dehydrated via a gradual series of ethanol solutions from 30 to 100%. Absolute ethanol was gradually substituted by a 1:1 solution of hexamethyldisilazane (HMDS)/absolute ethanol and successively by pure HMDS. Successively, HMDS was completely removed, and samples were dried in a desiccator. Dried samples were mounted on stubs, coated with gold (10 nm), and analyzed by a Cambridge 360 scanning electron microscope (Leica Microsystem, Wetzlar, Germany), as previously described. The number of cells with an altered shape was quantified by counting ≥ 500 cells (50 cells for each different SEM field with a magnification of 3000×) from triplicate samples.

### 4.4. Erythrocyte Deformability Measurement

The deformability of erythrocytes was determined by ektacytometry (LORRCA; Mechatronics Instruments BV, AN Zwaag, The Netherlands). The analytical method described by Donadello and collaborators [70] was used. The elongation index (EI) was obtained by the following equation: EI = (L−W)/(L + W), where L and W indicate the length and width of the diffraction pattern. A higher erythrocyte deformability corresponds to a higher EI for all shear stress values. We evaluated the EI curves for 12 values of shear stress. Since erythrocyte deformability achieves a plateau at 50 Pa, from these curves of shape modifications, the maximal elongation (EI max) has been calculated. The curves are represented in logarithmic scales.

### 4.5. Measurement of Na^+^/K^+^-ATPase Pump Activity

Na^+^/K^+^ ATPase activity was determined in healthy and β-Thal^+^ volunteers by measuring the release of inorganic phosphate (P_i_) from ATP [71,72]. Erythrocyte samples at 3% hematocrit were lysed in Tris-HCl (15 mM, pH 7.4) and centrifuged at 13,000× *g* for 15 min at 4 °C to remove hemoglobin. Briefly, 250 μL of distilled water was added to the erythrocyte membranes. The protein content of each sample was quantified by the Bradford method [73]. A 100 μL aliquot from each sample was incubated with 900 μL reaction buffer A (composition in mM: 50 Tris-HCl, 4 MgCl_2_, 3 ATP-Na_2_, pH 7.4) and reaction buffer B (composition in mM: 120 NaCl, 50 Tris-HCl, 20 KCl, 4 MgCl_2_, 3 ATP-Na_2_, pH 7.4) for 1 h at 37 °C. The reaction was stopped by adding 200 µL of 50% trichloroacetic acid (*v*/*v*). Then, ascorbic acid (2% *w*/*v*) was added for 20 min at room temperature. Absorbance was measured at 725 nm by spectrophotometry (Onda Spectrophotometer, UV-21, Carlsbad, California, United States). Na^+^/K^+^ ATPase activity was determined by subtracting the OD value of cuvette A from the OD value of cuvette B and expressed as P_i_/mg protein/h.

### 4.6. Assessment of Oxidative Stress Parameters

#### 4.6.1. Detection of Reactive Oxygen Species (ROS) Levels

The ROS levels were evaluated by the cell-permeable indicator 2′,7′-dichlorofluorescein diacetate (H2DCFDA, D6883, Sigma-Aldrich, Milan, Italy) in erythrocytes from healthy and β-Thal^+^ volunteers, according to the manufacturer’s instructions. As the positive control, human erythrocytes were incubated with 20 mM H_2_O_2_ at 25 °C for 30 min. ROS formation was determined by a microplate reader (Fluostar Omega, BMG Labtech, Ortenberg, Germany) with excitation and emission wavelengths of 485 nm and 535 nm, respectively. All readings were subtracted for the background fluorescence [74]. Results are expressed in arbitrary units.

#### 4.6.2. Measurement of Thiobarbituric-Acid-Reactive Substances (TBARS) Levels

Levels of thiobarbituric acid (TBA)-reactive substances (TBARS) were measured as reported by Mendanha and colleagues [75]. Human erythrocytes from healthy and β-Thal^+^ volunteers were suspended in isotonic solution (3% hematocrit, 100 µL), treated with 1.5 mL of 10% (*w*/*v*) acetic acid, and centrifuged (Neya 16R, 3000× *g*, 10 min). TBA (1%, 1 mL) was added to the supernatant, and the mixture was incubated at 95 °C for 30 min. Sample absorbance was measured at 532 nm (Onda spectrophotometer, UV-21, Carlsbad, CA, USA). Results are indicated as µM TBARS levels (1.56 × 10^5^ M^−1^ cm^−1^ molar extinction coefficient).

#### 4.6.3. Measurement of Total Sulfhydryl Groups (-SH) Content

The measurement of total -SH groups was performed as reported by Aksenov and Markesbery [76]. Human erythrocytes from healthy and β-Thal^+^ volunteers were centrifuged (Neya 16R, 1200× *g*, 5 min) and 8.5 µL of the pellet was hemolyzed in 1 mL of distilled water. A 20 μL aliquot of the hemolysis product was added to 940 µL of phosphate-buffered saline (PBS 0.1 M, pH 7.4) containing EDTA (1 mM). Then, the addition of 5,5′-dithiobis (2-nitrobenzoic acid) (DTNB, 50 mM, 30 μL) initiated the reaction, and the samples were incubated for 40 min at 25 °C protected from light. Control samples without cell lysate or DTNB were processed in parallel. After incubation, sample absorbance was measured at 412 nm (Onda spectrophotometer, UV-21, Carlsbad, CA, USA) and 3-thio-2-nitro-benzoic acid (TNB) levels were detected after subtraction of blank absorbance determined in samples containing DTNB only. To achieve full oxidation of -SH groups for the positive control, an aliquot of RBCs was incubated with 2 mM NEM for 1 h at 25 °C. Data were reported as μM TNB/mg protein.

#### 4.6.4. Measurement of Methemoglobin (MetHb) Content

The MetHb levels were determined as reported by Naoum and colleagues [77]. This assay is based on MetHb and (oxy)-hemoglobin (Hb) determination by spectrophotometry at 630 and 540 nm wavelength, respectively. In total, 100 μL of cell suspension obtained from healthy and β-Thal^+^ volunteers (3% hematocrit) was lysed in hypotonic buffer (15 mM NaH_2_PO_4_, 10 mM KH_2_PO_4_) containing 100 μL SDS 1% (hemolysis product A). A 300 μL aliquot of hemolysis product A was diluted with 3 mL hypotonic buffer (hemolysis product B). To induce complete Hb oxidation, a sample of cells was incubated for 1 h at 25 °C with 4 mM NaNO_2_, a well-known MetHb-forming compound. The absorbance of hemolysis product A and B samples was measured at 630 and 540 nm, respectively (Onda spectrophotometer, UV-21). The percentage (%) of MetHb was determined as follows: % MetHb = (OD 630 nm × 100)/(OD 630 nm + (OD 540 × 10)).

### 4.7. Measurement of Intracellular Free Iron

The intracellular free iron amount was determined using a commercially available colorimetric assay kit (cat. no. KT-757, Kamiya Biomedical Company, Seattle, WA, USA) in cells from healthy and β-Thal^+^ volunteers. In brief, dissociated iron is reduced and forms a chelate with the ferrozine chromogen. The absorbance of this colored complex is proportional to the iron concentration in the sample and is measured at 560 nm.

### 4.8. Preparation of Erythrocyte Plasma Membrane Proteins

Human erythrocytes from all healthy and β-Thal^+^ volunteers (3% hematocrit) were suspended in hypotonic cold solution (20 mM Tris-HCl, pH = 7.5) containing a protease inhibitor mixture and then centrifuged (Neya 16R, 13,000× *g*, 15 min, 4 °C) [78]. In contrast, plasma membrane proteins were processed as described by Pantaleo and collaborators [44]. In short, blood samples were suspended in a hypotonic cold solution containing a protease inhibitor mixture and centrifuged several times (Neya 16R, 13,000× *g*, 15 min, 4 °C) to discard hemoglobin, which remained in the supernatant. Next, the plasma membrane pellet was solubilized with SDS (1% *v*/*v*) and incubated on ice for 20 min. The solubilized plasma membrane proteins were stored at −80 °C.

#### SDS-PAGE Preparation and Western Blotting Analysis

Plasma membrane proteins were heated for 10 min at 95 °C after dissolving in Laemmli buffer [79]. The proteins were separated by SDS–polyacrylamide gel electrophoresis and transferred to a polyvinylidene fluoride membrane by maintaining a constant voltage for 2 h. Membranes were blocked for 1 h at 25 °C in BSA and incubated overnight at 4 °C with the primary antibodies diluted in TBST (mouse monoclonal anti-band 3 protein, B9277, and anti-pTyR (T1325), purchased by Sigma-Aldrich and diluted 1:5000 and 1:1000, respectively). Successively, membranes were incubated with peroxidase-conjugated goat anti-mouse IgG secondary antibodies (A9044, Sigma-Aldrich, Milan, Italy) diluted 1:10,000 in TBST solution for 1 h at 25 °C. To quantify the protein in equal amounts, a mouse monoclonal anti-β-actin antibody (A1978, Sigma-Aldrich, Milan, Italy, diluted 1:10,000) was incubated with the same membrane, as reported by Yeung and co-authors [80]. A system of chemiluminescence detection (Super Signal West Pico Chemiluminescent Substrate, Pierce Thermo Scientific, Rockford, IL, USA) was employed to obtain the signal for image analysis (Image Quant TL, v2003). The intensity of protein bands was determined by densitometry (Bio-Rad ChemiDocTM XRS+) [81].

### 4.9. Analytical Cytology

Human erythrocytes from healthy and β-Thal^+^ volunteers were fixed with 3.7% formaldehyde in PBS (pH 7.4) for 10 min at room temperature, washed in the same buffer, and permeabilized with 0.5% Triton X-100 (Sigma-Aldrich, Milan, Italy) in PBS for 5 min at room temperature. After washing with PBS, samples were incubated for 30 min at 37 °C with the following monoclonal antibodies: anti-band 3 (1:1000; Sigma-Aldrich, 136 B9277); anti-α/spectrin (1:1000; Santa Cruz Biotechnology, Dallas, TX, USA, sc-271130); anti-β/spectrin (1:1000; Santa Cruz Biotechnology, sc-374309); anti-band 4.1 (1:1000; Santa Cruz Biotechnology, sc-398983); and anti-α/actin (1:1000; Santa Cruz Biotechnology, sc-8432). Successively, all samples were washed thrice in PBS and then incubated for 30 min at 37 °C with anti-mouse IgG (H + L) Highly Cross-Adsorbed Secondary Antibody, Alexa Fluor™ 4888 (Invitrogen, Carlsbad, CA, USA) [82]. Samples were analyzed by an Olympus BX51 Microphot fluorescence microscope or by a FACScan flow cytometer (Becton Dickinson, Mountain View, CA, USA) equipped with a 488–544 nm argon laser. At least 20,000 events have been acquired. The median values of fluorescence intensity histograms are given to provide a semiquantitative analysis.

### 4.10. Measurement of SO_4_^2−^ Uptake

Band 3 protein anion exchange was determined as the uptake of SO_4_^2−^ in human erythrocytes from healthy and β-Thal^+^ volunteers, as previously reported [48,51,52,54,83,84,85,86]. Briefly, after washing, human erythrocytes were suspended in 35 mL SO_4_^2−^ medium (composition in mM: Na_2_SO_4_ 150, HEPES 5, glucose 5, pH 7.4, osmolarity 300 mOsm/kgH_2_O) and incubated at 25 °C for 5, 10, 15, 30, 45, 60, 90, and 120 min. After each incubation time, DIDS (10 μM), which is an inhibitor of band 3 activity [87], was added to 5 mL sample aliquots, which were kept on ice. To eliminate SO_4_^2−^ from the external medium, samples were washed three times in cold isotonic solution and centrifuged (Neya 16R, 4 °C, 1200× *g*, 5 min). Distilled water was added to the cell pellet to induce osmotic lysis, and perchloric acid (4% *v*/*v*) was used to precipitate proteins. After centrifugation (Neya 16R, 4 °C, 2500× *g*, 10 min), the supernatant, which contained SO_4_^2−^ trapped by erythrocytes during the fixed incubation times, was subjected to the turbidimetric analysis. To this end, 500 μL of the supernatant was sequentially mixed with 500 μL glycerol diluted in distilled water (1:1), 1 mL 4 M NaCl, and 500 μL 1.24 M BaCl_2_·2H_2_O. Finally, the absorbance of each sample was measured at 425 nm (Onda Spectrophotometer, UV-21). A calibrated standard curve, which was obtained in a separate experimental dataset by precipitating known SO_4_^2−^ concentrations, was used to convert the absorbance into [SO_4_^2−^] L cells × 10^−2^. The rate constant of SO_4_^2−^ uptake (min^−1^) was derived from the following equation: C_t_ = C_∞_ (1 − e^−rt^) + C_0_, where C_t_, C_∞_, and C_0_ indicate the intracellular SO_4_^2−^ concentrations measured at time t, ∞, and 0, respectively, e represents the Neper number (2.7182818), r indicates the rate constant of the process, and t is the specific time at which the SO_4_^2−^ concentration was measured. The rate constant is the inverse of the time needed to reach ~63% of total SO_4_^2−^ intracellular concentration [84]. Results are reported as [SO_4_^2−^] L cells × 10^−2^ and represent the SO_4_^2−^ micromolar concentration internalized by 5 mL erythrocytes suspended to 3% hematocrits.

### 4.11. Endogenous Antioxidant Activity Assessment

#### 4.11.1. Catalase Activity Assay

Catalase (CAT) activity was evaluated by the catalase assay kit (MAK381, Sigma-Aldrich, Milan, Italy) in cells obtained from healthy and β-Thal^+^ volunteers. As a positive control, cells were treated with 20 mM H_2_O_2_ for 30 min at 25 °C. Catalase activity was determined by reading the absorbance at 570 nm wavelength (Fluostar Omega, BMG Labtech, Ortenberg, Germany) after subtracting the background absorbance.

#### 4.11.2. Superoxide Dismutase Activity Assay

Superoxide dismutase (SOD) activity was evaluated by the SOD activity assay kit (CS0009, Sigma-Aldrich, Milan, Italy) in cells from healthy and β-Thal^+^ volunteers. The SOD enzyme was used to construct a standard curve of the linearized SOD rate. As a positive control, cells were treated with 20 mM H_2_O_2_ for 30 min at 25 °C. Superoxide dismutase activity was determined by reading the absorbance at a 450 nm wavelength (Fluostar Omega, BMG Labtech, Ortenberg, Germany) after subtracting the background absorbance.

#### 4.11.3. GSH/GSSG Ratio Measurement

GSH/GSSG ratio was quantified by the GSH assay kit (MAK440, Sigma-Aldrich, Milan, Italy) using an enzymatic recycling method with glutathione reductase in cells obtained from healthy and β-Thal^+^ volunteers. As a positive control, cells were treated with 2 mM NEM for 1 h at 25 °C. Sample absorbance was measured at 412 nm (Fluostar Omega, BMG Labtech, Ortenberg, Germany). Results are expressed as a GSH/GSSG ratio.

### 4.12. Experimental Data and Statistics

All data are expressed as the arithmetic mean ± SEM. For statistical analysis and graphics, GraphPad Prism (version 8.0, GraphPad software, San Diego, CA, USA) and Excel (Version 2019, Microsoft, Redmond, WA, USA) software were used. Significant differences between mean values were determined by an unpaired Student’s *t*-test or, alternatively, one-way or two-way analysis of variance (ANOVA), followed by Bonferroni’s multiple comparisons or Dunnett’s post-tests as appropriate, unless otherwise specified. Statistically significant differences were assumed at *p* < 0.05; n corresponds to the number of biological replicates.

## 5. Conclusions

In conclusion, the present study provides evidence regarding the downstream effects related to iron overload-induced oxidative stress at the level of structure and function in β-Thal^+^ erythrocytes. Specifically, band 3 protein hyperphosphorylation, clusterization, and the alteration of ion transport activity in β-Thal^+^ erythrocytes were shown for the first time. However, further research will help to determine whether such morphological and functional alterations could contribute to affecting systemic homeostasis in β-Thal^+^ subjects.

## Figures and Tables

**Figure 1 ijms-26-01593-f001:**
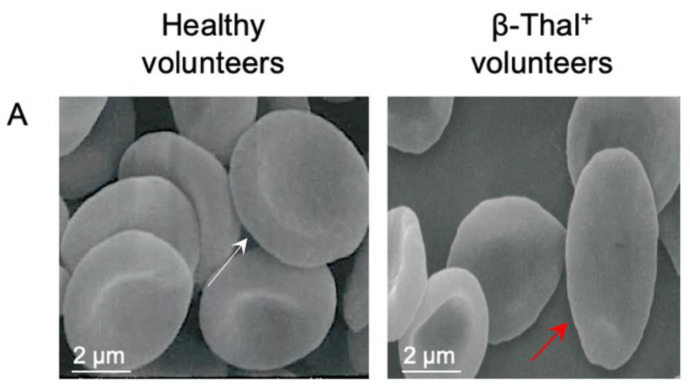
(**A**) Morphology evaluation in human erythrocytes obtained from healthy and β-Thal^+^ volunteers. SEM images show human erythrocytes with a canonical biconcave shape (white arrow) or, alternatively, elliptocytes (red arrows) in β-Thal^+^ volunteers. (**B**) Elongation index of erythrocytes obtained from healthy and β-Thal^+^ volunteers at different shear stress values (Pa). * *p* < 0.05 versus cells from healthy volunteers, unpaired Student’s *t*-test (*n* = 8).

**Figure 2 ijms-26-01593-f002:**
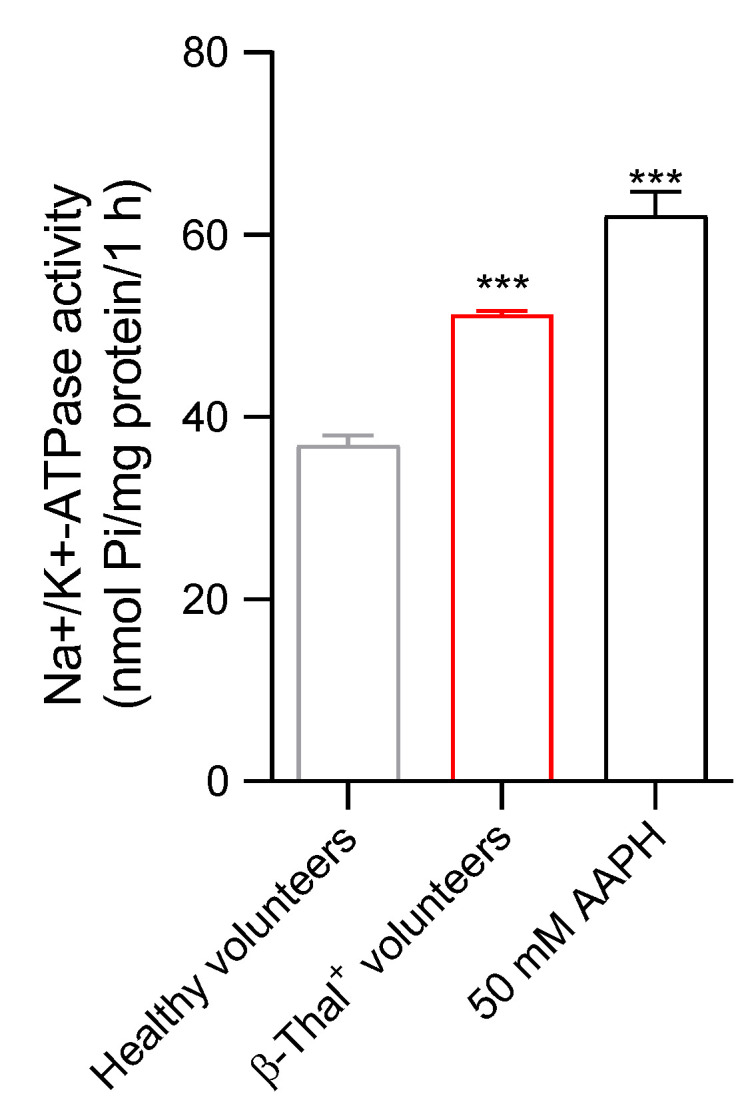
Detection of Na^+^/K^+^-ATPase pump activity in erythrocytes from healthy and β-Thal^+^ volunteers. *** *p* < 0.001 versus erythrocytes from healthy volunteers; one-way ANOVA followed by Bonferroni’s post-hoc test (*n* = 8).

**Figure 3 ijms-26-01593-f003:**
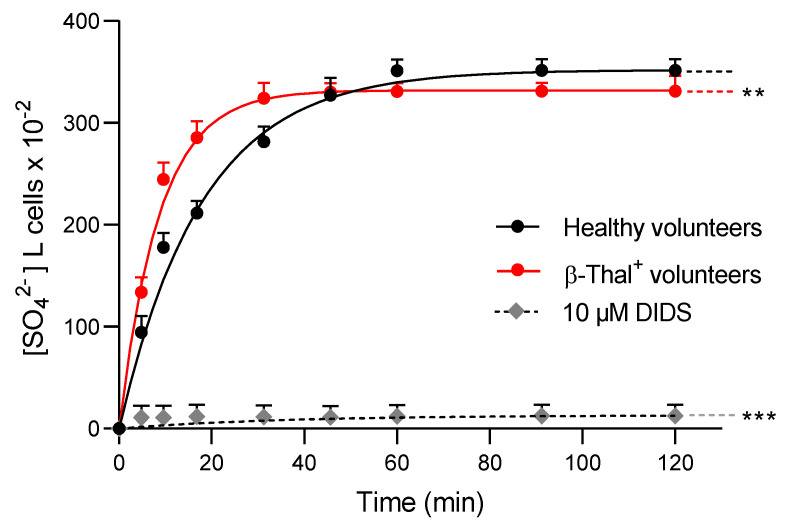
Time course of SO_4_^2−^ uptake in erythrocytes obtained from healthy and β-Thal^+^ volunteers. Erythrocytes obtained from healthy volunteers were also exposed to 10 µM DIDS in order to achieve complete inhibition of band 3 activity. ** *p* < 0.01 and *** *p* < 0.001 versus cells from healthy volunteers. Two-way ANOVA followed by Dunnett’s multiple comparison post-hoc test.

**Figure 4 ijms-26-01593-f004:**
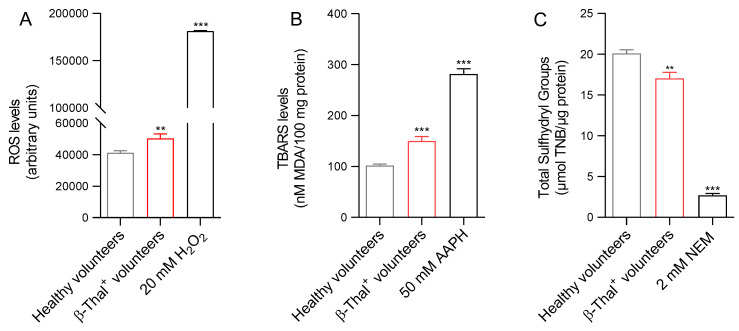
Evaluation of oxidative stress parameters. (**A**) Levels of intracellular ROS, (**B**) TBARS (µM MDA), and (**C**) content of total sulfhydryl groups (µM TNB/µg protein) from healthy and β-Thal^+^ volunteers. ** *p* < 0.01 and *** *p* < 0.001 versus erythrocytes from healthy volunteers, one-way ANOVA followed by Bonferroni’s post-hoc test (*n* = 15).

**Figure 5 ijms-26-01593-f005:**
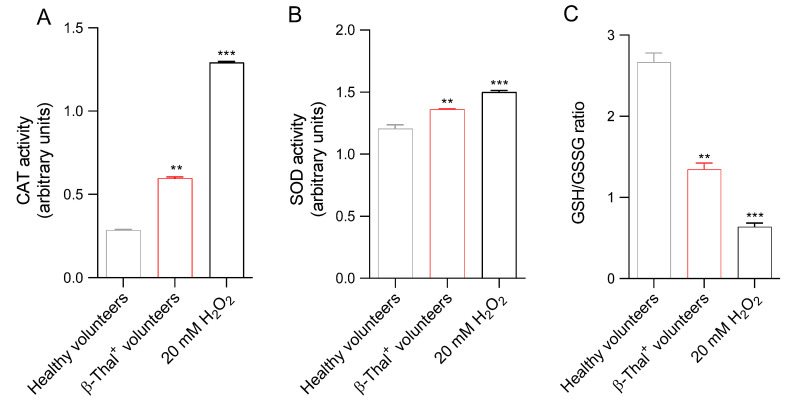
Assessment of the endogenous antioxidant capacity. (**A**) CAT activity, (**B**) SOD activity, and (**C**) GSH/GSSG ratio were detected in erythrocytes from healthy and β-Thal^+^ volunteers. Human erythrocytes from healthy volunteers were also exposed to 20 mM H_2_O_2_ (30 min at 25 °C). ** *p* < 0.01 and *** *p* < 0.001 versus cells from healthy volunteers, one-way ANOVA followed by Tukey’s post-hoc test (*n* = 10).

**Figure 6 ijms-26-01593-f006:**
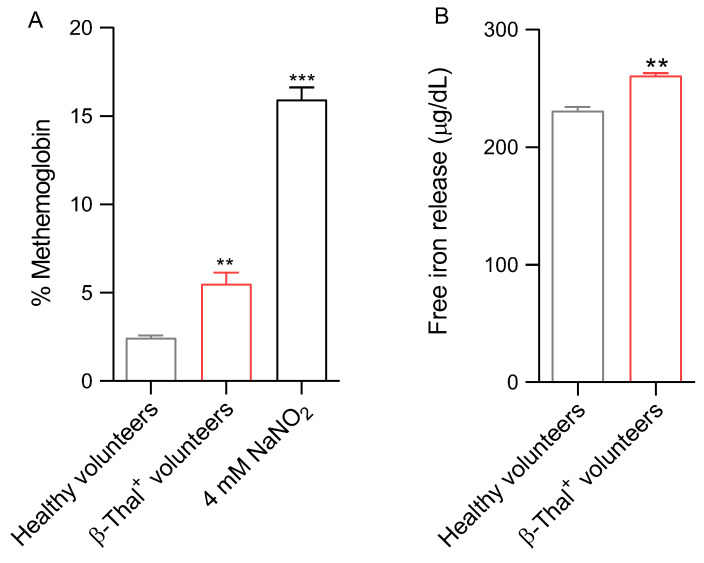
(**A**) Measurement of MetHb levels and (**B**) iron release in cells from healthy and β-Thal^+^ volunteers. ** *p* < 0.01 and *** *p* < 0.001 versus cells from healthy volunteers, one-way ANOVA followed by Bonferroni’s post-hoc test and paired Student’s *t*-test (*n* = 10).

**Figure 7 ijms-26-01593-f007:**
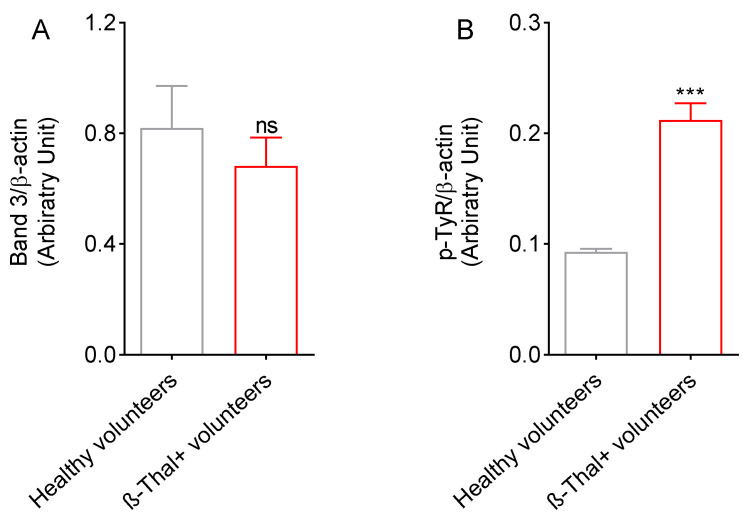
(**A**) Band 3 protein expression levels were detected in the plasma membranes of erythrocytes from healthy and β-Thal^+^ volunteers. (**B**) Band 3 protein phosphorylation levels were detected in the plasma membranes of erythrocytes from healthy and β-Thal^+^ volunteers. (*n* = 3). (**C**) Representative Western blotting images are shown. ns, not statistically significant versus erythrocytes from healthy volunteers; *** *p* < 0.001 versus erythrocytes from healthy volunteers, unpaired Student’s *t*-test.

**Figure 8 ijms-26-01593-f008:**
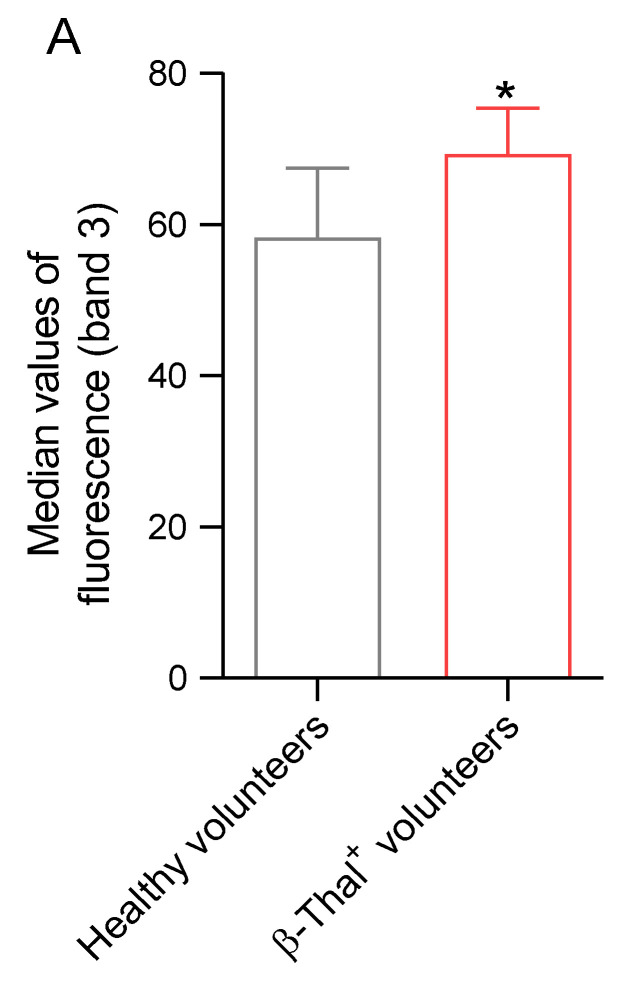
**Detection of band 3 protein distribution.** (**A**) Histograms reporting median values of band 3 fluorescence intensity detected in human erythrocytes from healthy and β-Thal^+^ volunteers. (**B**) Representative images of immunofluorescence showing band 3 protein distribution. Morphological and distribution changes are indicated by red arrows. The dotted yellow perimeter identifies an elliptocyte. Samples were observed with a 100× objective. * *p* < 0.05 versus healthy volunteers, unpaired Student’s *t*-test (*n* = 6).

**Figure 9 ijms-26-01593-f009:**
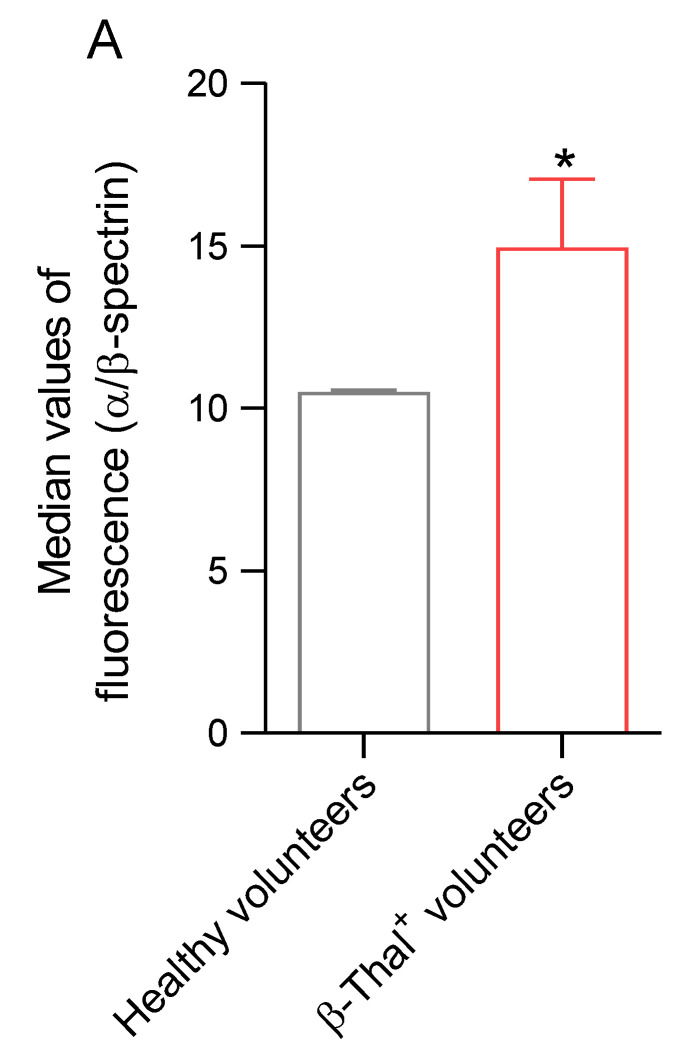
**Detection of α/β-spectrin distribution.** (**A**) Histograms reporting median values of protein fluorescence intensity detected in human erythrocytes from healthy and β-Thal^+^ volunteers. (**B**) Representative images of immunofluorescence showing the distribution of both proteins. Distribution changes (aggregates) are indicated by red arrows. Samples were observed with a 100× objective. * *p* < 0.05 versus healthy volunteers, unpaired Student’s *t*-test (*n* = 6).

**Figure 10 ijms-26-01593-f010:**
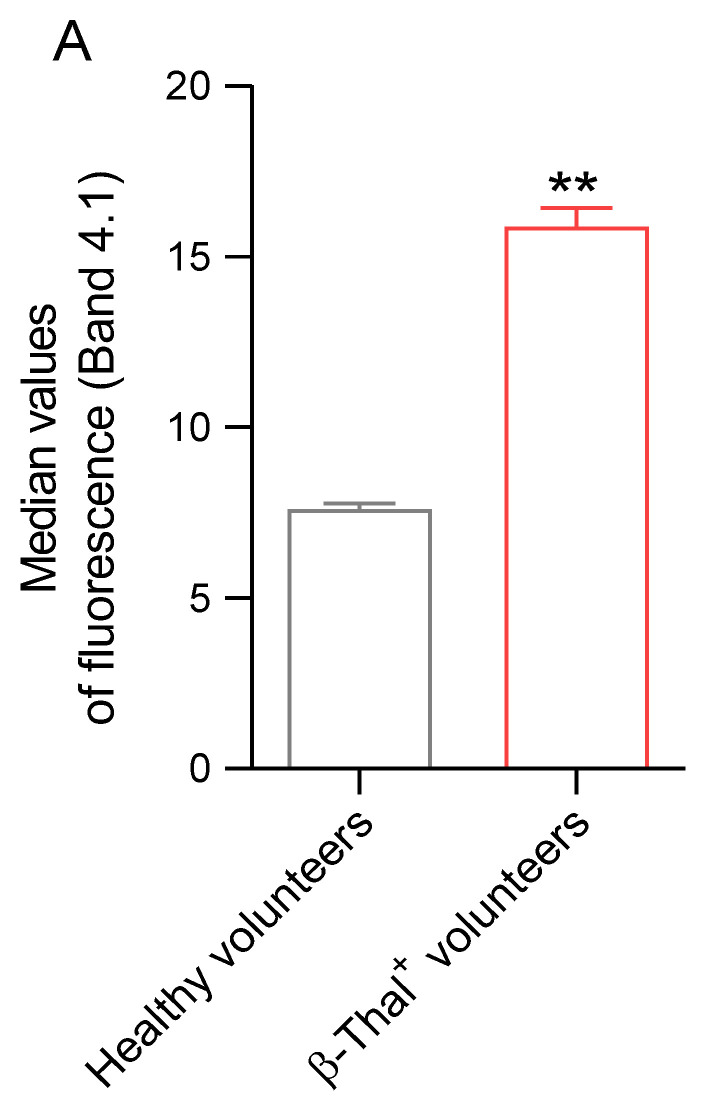
**Detection of band 4.1 distribution.** (**A**) Histograms reporting median values of protein fluorescence intensity detected in human erythrocytes from healthy and β-Thal^+^ volunteers. (**B**) Representative images of immunofluorescence showing the distribution of band 4.1. Distribution changes are indicated by red arrows. Samples were observed with a 100× objective. ** *p* < 0.05 versus healthy volunteers, unpaired Student’s *t*-test (*n* = 6).

**Figure 11 ijms-26-01593-f011:**
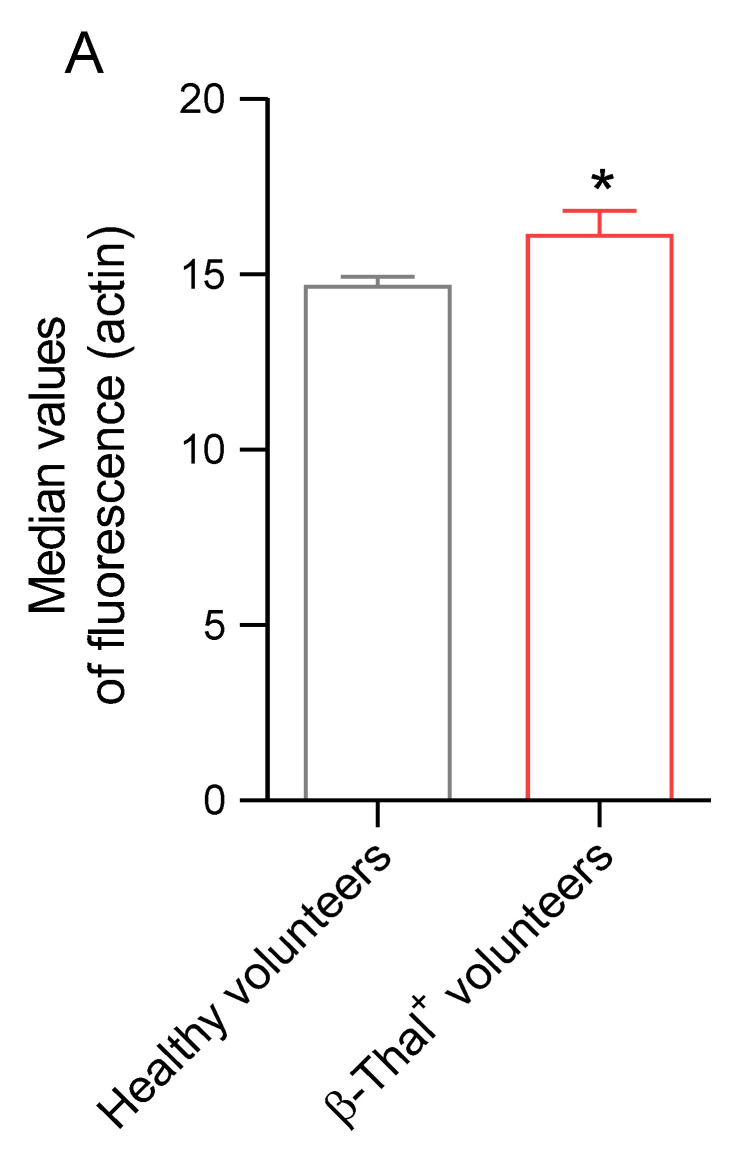
**Detection of α-actin distribution.** (**A**) Histograms reporting median values of protein fluorescence intensity detected in human erythrocytes from healthy and β-Thal^+^ volunteers. (**B**) Representative images of static cytometry showing protein distribution. Distribution changes (clusters) are indicated by arrows. Samples were observed with a 100× objective. * *p* < 0.05 versus healthy volunteers, unpaired Student’s *t*-test (*n* = 6).

**Table 1 ijms-26-01593-t001:** Rate constant of SO_4_^2−^ uptake and SO_4_^2−^ amount trapped in human erythrocytes from healthy and β-Thal^+^ volunteers. Results are presented as the mean ± SEM from (*n*) biological replicates. ns, not statistically significant versus healthy volunteers; ** *p* < 0.01 and *** *p* < 0.001 versus healthy volunteers. Two-way ANOVA followed by Dunnett’s multiple comparison post-hoc test.

Experimental Condition	Rate Constant (min^−1^)	Time (min)	*n*	SO_4_^2−^ Amount Trapped After 45 min of Incubation in SO_4_^2−^ Medium [SO_4_^2−^] l cells ×10^2−^
Healthy volunteers	0.056 ± 0.010	17.79	10	327.33 ± 9.67
β-Thal^+^ volunteers	0.115 ± 0.021 **	8.62	10	330.00 ± 11.31 ^ns^
10 µM DIDS	0.023 ± 0.002 ***	41.97	10	16.5 ± 0.51 ***

**Table 2 ijms-26-01593-t002:** Hematological parameters detected in healthy and β-Thal^+^ volunteers, according to gender. Data are presented as the mean ± standard deviation. Hb (Hemoglobin), MCH (mean corpuscular hemoglobin), MCHC (mean corpuscular hemoglobin concentration), MCV (mean corpuscular volume), Hct (hematocrit).

Sex	Volunteers	Hb (g/dL)	MCH (pg)	MCHC (g/dL)	MCV (fL)	RBC Count (10³/µL)	Hct (%)
Male							
*n* = 28	Healthy volunteers	13.9 ± 1.5	31.2 ± 3	32.8 ± 0.7	95.6 ± 1.1	4.5 ± 0.9	42.7 ± 5.8
*n* = 21	β-Thal^+^ volunteers	10.5 ± 1.7	22.1 ± 3.1	30.9 ± 1.3	71.6 ± 8.7	5.2 ± 1	37.3 ± 2.6
Female							
*n* = 25	Healthy volunteers	13.5 ± 0.7	29.8 ± 1.6	32.9 ± 0.8	91.5 ± 3.2	4.5 ± 0.31	40.8 ± 3.4
*n* = 23	β-Thal^+^ volunteers	9.4 ± 2.1	20.7 ± 3.9	30.3 ± 1.8	68.3 ± 8	5.3 ± 0.8	34.3 ± 4.3

## Data Availability

The data that support the findings of this study are available from the corresponding author upon reasonable request.

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
