# Peer review of "Iron Overload-Related Oxidative Stress Leads to Hyperphosphorylation and Altered Anion Exchanger 1 (Band 3) Function in Erythrocytes from Subjects with β-Thalassemia Minor"

_ijms, 2025, doi:10.3390/ijms26041593_

Round 1

Reviewer 1 Report

Comments and Suggestions for Authors

This work is very thorough and meticulous. The authors have gathered a lot of data, characterizing almost every aspect of RBCs. Moreover, I really enjoyed reading the Discussion section that contains so much information and beautifully connects the findings. I have some comments that I would like to be addressed before the manuscript final publication.

Major comments:

1) I believe that the results would be easier to follow if stratified in more general sections instead of having a subtitle for every single assay. E.g., the authors could have a section regarding shape and plasticity, another one for all redox parameters (as they already do), another one for pump/channel activity (for ATPase and Band 3), another one for membrane/cytoskeletal protein distribution and levels etc. In this way, the reader could better grasp the whole picture of the findings and connect similar or complementary results together.

2) Since the findings are many and are really important, I would suggest for a graphical figure to be included in the end, to combine all findings that characterize beta-thalassemia trait RBCs. This could enhance the  work's impact by making the results more clear to the reader.

Minor comments:

1) In the Methods section, the title "Biological samples and blood unit preparation" implies the examination of blood units. Nonetheless, the authors studied freshly drawn blood, therefore this title should be altered to better reflect the study.

2) Regarding some of the study's findings, there is published work that is important to be acknowledged (e.g., for protein-related data, like tyrosine phsophorylation, spectrin etc there is a proteomic work in beta-thalassemia minor 10.3390/ijms22073369). Please check for other relevant publications that may be missing.

Author Response

  • Reviewer 1

This work is very thorough and meticulous. The authors have gathered a lot of data, characterizing almost every aspect of RBCs. Moreover, I really enjoyed reading the Discussion section that contains so much information and beautifully connects the findings. I have some comments that I would like to be addressed before the manuscript final publication.

We thank the Reviewer for the overall positive evaluation.

Major comments:

I believe that the results would be easier to follow if stratified in more general sections instead of having a subtitle for every single assay. E.g., the authors could have a section regarding shape and plasticity, another one for all redox parameters (as they already do), another one for pump/channel activity (for ATPase and Band 3), another one for membrane/cytoskeletal protein distribution and levels etc. In this way, the reader could better grasp the whole picture of the findings and connect similar or complementary results together.

We want to thank the Reviewer for raising this point. As suggested, the result section was now re-arranged into the following chapters: 2.1 β-Thal+ erythrocyte shape and plasticity; 2.2 Evaluation of ion transport in β-Thal+ erythrocytes, including 2.2.1 Na+/K+ ATPase activity, and 2.2.2 SO42- uptake via band 3 protein; 2.3 Oxidative stress in β-Thal+ erythrocytes, divided into four subchapters including the endogenous antioxidant capacity; 2.4 Detection of MetHb levels and intracellular iron release in β-Thal+ erythrocytes; 2.5 Assessment of membrane and cytoskeletal protein levels and distribution in β-Thal+ erythrocytes, divided into four subchapters. We acknowledge that these modifications improved the logical flow of the Results section.

2) Since the findings are many and are really important, I would suggest for a graphical figure to be included in the end, to combine all findings that characterize beta-thalassemia trait RBCs. This could enhance the work's impact by making the results clearer to the reader.

We want to thank the Reviewer for her/his suggestion. A graphical abstract has been added to the manuscript. This is compliant with the journal´s requirements.

Minor comments:

In the Methods section, the title "Biological samples and blood unit preparation" implies the examination of blood units. Nonetheless, the authors studied freshly drawn blood, therefore this title should be altered to better reflect the study. Done.

2) Regarding some of the study's findings, there is published work that is important to be acknowledged (e.g., for protein-related data, like tyrosine phosphorylation, spectrin etc there is a proteomic work in beta-thalassemia minor 10.3390/ijms22073369). Please check for other relevant publications that may be missing.

We want to thank the Reviewer for her/his suggestion. Done.

Reviewer 2 Report

Comments and Suggestions for Authors

In the paper “Iron overload-related oxidative stress impairs membrane de-2 formability of erythrocytes from subjects with β-thalassemia 3 minor: role of anion exchanger 1 (AE1) and Na+-K+/ATPase ac-4 tivity” In this paper, authors describe the structural and functional changes in erythrocytes from β-Thal+ subjects due to increased oxidative stress. The study found significant changes in cell shape, including a higher number of elliptocytes, and iron overload that led to increased production of reactive oxygen species (ROS). This resulted in lipid peroxidation, protein oxidation, and methemoglobin formation, which impaired Na+/K+-ATPase activity and erythrocyte deformability. Additionally, alterations in cytoskeletal proteins and band 3 ion transport were observed, along with overactivation of antioxidant enzymes and glutathione depletion. Authors used variety of methods to determined blood indicators that directly and indirectly are related to oxidative stress in β-Thalassemia minor patients. However, the hypothesis is not clear and, in general, the observed changes have been already extensively described in various articles related to oxidative stress and β-Thalassemia.

-          Remove from the paper figure 4B , you cannot measure the free iron in the blood/ serum (it is not stated clear in the Materials and method section) in samples that were collected to tubes containing EDTA – iron chelator. It is clearly stated in the protocol of the kit that you used: “SAMPLE PREPARATION 1. Serum or plasma -  Insoluble substances in serum and plasma samples should be removed by filtration or centrifugation. EDTA-plasma samples cannot be used as EDTA interferes with this assay”

-          You didn’t use lines that indicate statistical significance in the graphs. I understand that used asterisks indicate significant differences between the control group and the studied groups of patients? You only used this line once in Figure 2, please be consistent.

-          What for you present results for the “third group” that show artificial conditions? They could, in some way, be just confirmation of the methodology used, but they do not provide much additional information in terms of differences between healthy individuals and those suffering from β-thalassemia.

-          In figure 5 you present results of western blot analysis, why you performed only analysis on 3 + 3 samples?

-          Describing figures 8, 9 and 10, you use the same sentence: “shows an intense rearrangement and redistribution (red arrows)”. Expand this description, what changes in location and quantity were observed? These figures could be smaller - reduced graph next to picture.

Author Response

  • Reviewer 2

In the paper “Iron overload-related oxidative stress impairs membrane de-2 formability of erythrocytes from subjects with β-thalassemia 3 minor: role of anion exchanger 1 (AE1) and Na+-K+/ATPase ac-4 tivity” In this paper, authors describe the structural and functional changes in erythrocytes from β-Thal+ subjects due to increased oxidative stress. The study found significant changes in cell shape, including a higher number of elliptocytes, and iron overload that led to increased production of reactive oxygen species (ROS). This resulted in lipid peroxidation, protein oxidation, and methemoglobin formation, which impaired Na+/K+-ATPase activity and erythrocyte deformability. Additionally, alterations in cytoskeletal proteins and band 3 ion transport were observed, along with overactivation of antioxidant enzymes and glutathione depletion. Authors used variety of methods to determined blood indicators that directly and indirectly are related to oxidative stress in β-Thalassemia minor patients. However, the hypothesis is not clear and, in general, the observed changes have been already extensively described in various articles related to oxidative stress and β-Thalassemia.

We would like to thank the Reviewer for the time taken to revise our manuscript. A PubMed search with the keywords (β-Thalassemia minor), (oxidative stress) and (human erythrocytes) restitutes about 10 results. Therefore, it appears that the relationship between the increased oxidative stress in β-Thalassemia minor (β-Thal+) erythrocytes and their functional properties is still poorly documented. In particular, we hypothesized that increased oxidative stress, potentially provoked by an iron overload, may lead to the dysfunction (structure and functional) of anion exchanger 1 (band 3) in erythrocytes from β-Thal+ volunteers. This hypothesis has now been better specified in the abstract and introduction. The novelty of our findings has now been better emphasized in a new manuscript title and in the revised abstract and conclusions.

-Remove from the paper figure 4B, you cannot measure the free iron in the blood/ serum (it is not stated clear in the Materials and method section) in samples that were collected to tubes containing EDTA – iron chelator. It is clearly stated in the protocol of the kit that you used: “SAMPLE PREPARATION 1. Serum or plasma-Insoluble substances in serum and plasma samples should be removed by filtration or centrifugation. EDTA-plasma samples cannot be used as EDTA interferes with this assay”.

We want to thank the Reviewer for raising this point. In the data sheet the following information is reported:

SAMPLE PREPARATION: tissue extracts, cell lysates, and other samples such as urine or other biological fluids. In fact, we performed the measure of free iron in cell (erythrocyte) lysates.

In the section of Materials and Methods, we reported that human erythrocytes were washed in isotonic solution (NaCl 150 mM, 4-(2-hydroxyethyl)-1-piperazineethanesulfonic acid (HEPES) 5 mM, glucose 5 mM, pH 7.4, osmolarity 300 mOsm/kgH2O) and centrifuged (Neya 16R, 1200×g, 5 min) to discard plasma and buffy coat. Finally, human erythrocytes were suspended in isotonic solution according to the experimental tests described below. After washing, EDTA is removed, and cannot interfere with iron assay. In particular, we detected the intracellular free iron content. Based on what reported, degradation of the unstable hemoglobin and iron overload represent the leading causes of oxidative damage in erythrocyte cells from β-Thal+ subjects. In the full text, "free iron" has been modified in "intracellular free iron".

-You didn’t use lines that indicate statistical significance in the graphs. I understand that used asterisks indicate significant differences between the control group and the studied groups of patients? You only used this line once in Figure 2, please be consistent.

We want to thank the Reviewer for raising this point. Done.

-What for your present results for the “third group” that show artificial conditions? They could, in some way, be just confirmation of the methodology used, but they do not provide much additional information in terms of differences between healthy individuals and those suffering from β-thalassemia.

We want to thank the Reviewer for raising this point. As the Reviewer points out, a positive control has been introduced for each assay, which is fundamental to validate the specific methodology and was not meant (and nowhere in the manuscript is stated that it is meant) to recapitulate the β-thalassemia conditions. All the information related to positive controls is detailed in the sections of both Materials and Methods and Results.

-In figure 5 your present results of western blot analysis, why you performed only analysis on 3 + 3 samples?

We want to thank the Reviewer for raising this point. In this study, 44 volunteers (21 males and 23 females) with β-thalassemic minor (β-Thal+) and 53 healthy volunteers (28 males and 25 females) were enrolled. n=3+3 is a biological triplicate, obtained from an erythrocyte pool belonging to 53 healthy volunteers, or alternatively, 44 volunteers with β-thalassemic minor.

In the section Materials and Methods, the preparation of erythrocyte plasma membrane proteins is reported. Human erythrocytes from all healthy and β-Thal+ volunteers (3% hematocrit) were suspended in hypotonic cold solution (20 mM Tris-HCl, pH=7.5) containing a protease inhibitor mixture and then centrifuged (Neya 16R, 13000×g, 15 min, 4 °C) [1,2]. Instead, plasma membrane proteins were processed as described by Pantaleo and collaborators [3]. In short, blood samples were suspended in a hypotonic cold solution containing a protease inhibitor mixture and centrifuged several times (Neya 16R, 13000 ×g, 15 min, 4 °C) to discard hemoglobin, which remained in the supernatant. Next, the plasma membrane pellet was solubilized with SDS (1% v/v) and incubated on ice for 20 min. The solubilized plasma membrane proteins were stored at -80 °C.

-Describing figures 8, 9 and 10, you use the same sentence: “shows an intense rearrangement and redistribution (red arrows)”. Expand this description, what changes in location and quantity were observed?

We want to thank the Reviewer for raising this point. These rearrangements are now better described in the Results section.

These figures could be smaller -reduced graph next to picture. Done.

  1. Vona, R.; Gambardella, L.; Cittadini, C.; Straface, E.; Pietraforte, D. Biomarkers of Oxidative Stress in Metabolic Syndrome and Associated Diseases. Oxid Med Cell Longev 2019, 2019, 8267234, doi:10.1155/2019/8267234.
  2. Vona, R.; Gambardella, L.; Ortona, E.; Santulli, M.; Malorni, W.; Care, A.; Pietraforte, D.; Straface, E. Functional Estrogen Receptors of Red Blood Cells. Do They Influence Intracellular Signaling? Cell Physiol Biochem 2019, 53, 186-199, doi:10.33594/000000129.
  3. Pantaleo, A.; Ferru, E.; Pau, M.C.; Khadjavi, A.; Mandili, G.; Matte, A.; Spano, A.; De Franceschi, L.; Pippia, P.; Turrini, F. Band 3 Erythrocyte Membrane Protein Acts as Redox Stress Sensor Leading to Its Phosphorylation by p (72) Syk. Oxid Med Cell Longev 2016, 2016, 6051093, doi:10.1155/2016/6051093.

Round 2

Reviewer 1 Report

Comments and Suggestions for Authors

The authors have answered all my comments and I believe their manuscript can be accepted in its current form.

Reviewer 2 Report

Comments and Suggestions for Authors

The authors responded to all my comments. I recommend the manuscript for acceptance.